# 3D Scanner-Based Identification of Welding Defects—Clustering the Results of Point Cloud Alignment

**DOI:** 10.3390/s23052503

**Published:** 2023-02-23

**Authors:** János Hegedűs-Kuti, József Szőlősi, Dániel Varga, János Abonyi, Mátyás Andó, Tamás Ruppert

**Affiliations:** 1Faculty of Informatics, Savaria Institute of Technology, Eotvos Lorand University, H-9700 Szombathely, Hungary; 2ELKH-PE Complex Systems Monitoring Research Group, Department of Process Engineering, University of Pannonia, H-8200 Veszprem, Hungary

**Keywords:** welding, seam defect recognition, DBSCAN, cloud points matching

## Abstract

This paper describes a framework for detecting welding errors using 3D scanner data. The proposed approach employs density-based clustering to compare point clouds and identify deviations. The discovered clusters are then classified according to standard welding fault classes. Six welding deviations defined in the ISO 5817:2014 standard were evaluated. All defects were represented through CAD models, and the method was able to detect five of these deviations. The results demonstrate that the errors can be effectively identified and grouped according to the location of the different points in the error clusters. However, the method cannot separate crack-related defects as a distinct cluster.

## 1. Introduction

Industrial digitalization has opened up new dimensions to make industrial processes more efficient. The quality control of the welding is a significant challenge to achieve autonomous manufacturing [1]. The issue of compliance and the existence of quality has become increasingly prominent with the concept of quality management [2]. Various machine learning (ML) techniques can be used to solve problems related to welding processes and make manufacturing more efficient [3]. The accuracy of welding processes improved significantly with ML algorithms [4].

Welding technology encompasses a wide range of processes and variations, making it a subject of extensive research. The use of raw materials in welding has been studied in depth, with research focusing on the structural changes in the materials used for joining [5]. This research typically includes laboratory testing with both destructive and non-destructive methods, as well as microscopic examinations [6,7]. Additionally, there is ongoing research on seam detection using image processing algorithms (based on laser vision) and Kalman filter for trajectory optimization [8]. The laser vision technique [9] is also used for adaptive measuring the actual weldments. The algorithm extracts the shape features by constructing the characteristic triangle of the weld trajectory [10]. In recent years, there has been an increase in research on artificial intelligence (AI) in the field of welding, which represents a promising new direction for research [11]. Breakthroughs in this area, as well as related areas, are expected in the near future [12,13].

The test to verify the conformity of a welded joint is complex, due to the thermal processes used in the technology. Standards, process safety, and economic aspects are important factors to be incorporated. The test order is standardized. The listing surface and volume deviations using defect categories and codes are one of the significant achievements in welding quality management [2]. The standard shows three quality categories with limits for each deviation. On this basis, clear definitions are introduced for the acceptability of welded joints. A welded joint is considered to be in compliance if the extent of any defects it may contain is within the limits; otherwise, it is considered to be non-compliant. In some cases, such as cracks, they are not acceptable in either quality class (*not permitted*) [1]. The physical principle of the various non-contact inspection tools, successfully used in many industrial applications to detect quality defects, belongs to the research of image recognition and image processing [14]. Image processing tools are used for 3D reconstruction, for example, in medical applications [15] and in specialized fields such as identification of underwater objects [16].

The question of inhomogeneity of welds is also a challenge for non-destructive testing using ultrasonic waves, which is applied in power plants to inspect welds. Among different ultrasonic testing techniques, phased array ultrasonic testing (PAUT) determines the angle and focal law and transmits pulses through the electromagnetic time delays of multiple elements arrayed in phased array (PA) probes to generate a focused beam. It can detect flaws present in multiple directions of the target under inspection [17].

Three-dimensional scanning is an accurate and fast non-contact measurement process that creates a 3D point cloud of physical objects. Active non-contact scanning methods can work using time of flight, triangulation, or structured light. The first two are specific to laser scanning [18]. This process emits laser light or infrared or other special colored light and then uses the returned light to create a spatial point mesh as a digital model by capturing the surface coordinates (*x*, *y*, *z*) of the actual object (up to millions of points per second). The technology is often used for reverse engineering, which creates a digital duplicate of reality [19]. Technology is becoming increasingly accurate to meet design challenges. The optics measure both the distortion of the grating and the intensity of the reflected light, giving similarly accurate results as laser versions [20]. The accuracy of five microns experienced in 3D scanning processes [21] is strongly setting-dependent (scanning conditions, part surface quality). A laser scanning test has been considered for future work to achieve similar accuracy, as 3D visualization models of complex and irregular schemes can be constructed quickly and accurately [22]. These scanners have fixed and handheld peripherals that can be adapted in the laboratory or to a robot. Increasingly sophisticated equipment has enabled advanced point cloud processing (after scanning of parts) and faster non-destructive material testing (comparison).

The point clouds should be sampled for more efficient data processing. One possible solution to the processing of point clouds is fast global registration (FGR) and the iterative closest point (ICP) algorithm, which creates pairwise correspondences between two point clouds [23]. After fitting the point clouds, the distances can be calculated, and the deviation can be identified with density-based clustering [24]. Density-based spatial clustering of applications with noise (DBSCAN) is the pioneer of density-based clustering techniques [23], which can discover clusters of arbitrary shape and effectively handle noise or outliers. The DBSCAN algorithm has a quadratic time complexity with dataset size and can be extended to large datasets [25]. DBSCAN allows the identification of clusters of different shapes and can handle noisy patterns in the data by classifying points in low-density regions as noise and skipping them. Its operation is based on spatial density, which has several advantages over other classical clustering algorithms. One of these advantages is its ability to identify clusters of arbitrary shapes correctly. Another is that it performs divisive clustering by automatically determining the number of clusters, so it is not necessary to predetermine their number.

The novelties and objectives of this article are as follows:A comprehensive discussion of the classification of weld defects is made based on image processing and data analysis, including the qualitative variation of each weld defect;A methodology is developed for the visualization of weld defects using computer-aided design (CAD) with a results-oriented approach;The point clouds processed by the 3D camera and modeled with CAD programs are determined (point feature histogram—PFH); the descriptive characteristics of the points prepares the clouds for coarse (FGR) and fine (ICP) matching;The weld defects are detected by density-based clustering (DBSCAN) by sorting the top distance between point pairs, and the clustered errors are aligned to the standard.

The paper is structured as follows. The developed framework is detailed in Section 2. In Section 2.1, the modeling of the defective CAD parts and the reference CAD workpiece are presented, converting them into a point cloud, and then calculating point properties (Section 2.2). The fitting process is performed using FGR and ICP algorithms (Section 2.3). The clustering process using the DBSCAN clustering algorithm is shown in Section 2.4. Finally, the results of the developed framework are presented in Section 3.

## 2. The Proposed CAD Model-Based Method of Generation and Analysis of Error Distributions

A clustering-based framework is developed to detect qualitative variations of welded joint defects (possible geometric variations of joints, weld convexity, root size smaller or larger, edge efflorescence, inclusions, end crater, and cracks). The proposed framework (see in Figure 1) started with the creation of point clouds. For the fitting, we need two point clouds: a reference point cloud, which is a point cloud of a flawless part, and a test point cloud, which is a point cloud of a defective part (*Input* block on Figure 1). The process continues with sampling. After determining the point descriptor properties, a coarse and fine-fitting process can follow (*Cloud points matching* block on Figure 1). Finally, the clustering step localizes the centers of the errors (*Clustering* block on Figure 1).

A 3D reconstruction of the object was performed using structured light 3D scanning. The resulting high-density point cloud was then converted into a mesh 3D model. An initial step was added to the testing procedure to account for potential defects in real weldments. CAD models of fillet welded models were created, incorporating non-conformities as outlined in ISO 5817:2014. These non-conformities were grouped into three quality categories, with specific limits defined for each. For example, imperfection 5213, with a deviation larger than the specified root size, was included as a separate CAD model. The point clouds were compared from both the compliant and non-compliant CAD models, eliminating other uncertainties present in real parts [26]. In other words, the system is trained with CAD models, and its operation is validated through a 3D point cloud of a real welded joint. These point clouds with errors are compared one by one with the point cloud of the CAD model without errors. The purpose of the analysis was to obtain the deviations associated with each error from the point cloud comparison and to explore their correlations.

The steps are described in the following subsections:The prepared welded parts are scanned with the 3D scanner. CAD parts with defects and a reference CAD workpiece are modeled (see *Input* block on Figure 1) and converted into a point cloud for the training and test set (Section 2.1).Sampling generates a point cloud from the models (mesh files) for the validation set (see *Cloud points matching* block on Figure 1). The point features should be calculated to compare the two point clouds (Section 2.2).After computing the point properties, the technique of matching the point clouds using the FGR and ICP algorithms (see *Cloud points matching* block on Figure 1) is described (Section 2.3).The optimal percentile value is selected for PFH to facilitate efficient clustering (see *Clustering* block on Figure 1). DBSCAN clustering is applied (Section 2.4).

### 2.1. Training and Validation Datasets

In the next step, point clouds of the different CAD models are created to compare the real weldments. The following mathematical basis is used to compare the point clouds:*Modeling the CAD models:*$\hat{P}$ is the 3D point-cloud of the masterpiece with ${\hat{p}}_{i}$ ($i=1\ldots N_{\hat{p}}$) 3D points, where $N_{\hat{p}}$ is the number of points; $\hat{Q}$ is the 3D point-cloud of the test piece with ${\hat{q}}_{j}$ ($j=1\ldots N_{\hat{q}}$), where $N_{\hat{q}}$ is the point number in the $\hat{Q}$ cloud. For optimal point matching, sampling is required.*Sampling the resulting point clouds:* Sampling is the systematic selection of a relatively small number of representative elements or sub-clouds. After sampling, the point clouds $P=\left\{p_{1},\ldots ,p_{N}\right\}$ and $Q=\left\{q_{1},\ldots ,q_{N}\right\}$ consisted of the same number (N) of points necessary for a proper fit.

The scanning of the workpiece (Figure 2) for the validation set consists of the following steps:1.Tray calibration according to the size of the workpiece. The projection grid is projected by the projector onto the surface of the tray, followed by scanning of other details of the background (rotation table).2.The flawless workpiece is placed in the center of the tray, adjusting the rotation angle, and the scanning process begins. The three-dimensional model, usually fused from six or seven images, can be exported as a mesh model in various (*.stl*/*.obj*) formats.3.Since the scanned file is currently a mesh model consisting of 3D points and triangles (called surfaces), a point cloud is created from the models and used as the experimental subject (data source) for the study.4.The process is repeated with the workpiece containing the weld defect, with recommended setup.

### 2.2. Determination of the Point Features

In the next step, the point feature descriptors (fast point feature histogram—FPFH) [27] in P and Q are calculated (see Figure 3). According query points are $p_{s}$ to $p_{i}$ and $q_{s}$ to $q_{i}$ neighbors. After calculating the point descriptors, point correspondences can be established by nearest-neighbor feature matching. The FGR filters the results with multiple tests (correspondence rejection) to exclude false correspondences. It then estimates a transformation to align the two clouds using the remaining correspondences. This transformation is the initial transformation for the next step (transformation refinement with ICP).

Our assumption is that the two objects are roughly identical, but due to the welding errors, there are differences between the ideal and the point cloud at the welding area. Therefore, we intended to match the two point clouds by performing a rough feature-based registration. For this, point feature descriptors must compute the points in both clouds. A point descriptor is a high-dimensional vector that encapsulates information and describes the underlying surface at a query point. There are many feature descriptor algorithms. We apply the FPFH [28], following the authors of FGR. The FPFH can compute efficiently and provides good matching accuracy.

The FPFH is based on the PFH, which was an earlier and slightly different version with higher computational complexity and dimensionality. Both of them represent the relationship (Equation (Equation 1)) between normal vectors of point pairs ($n_{s}$, $n_{t}$) using different angular features (α, β, γ). A point pair ($p_{1}$, $p_{s}$) needs to first define (Equation (Equation 2)) a fixed coordinate frame (**uwv**) to calculate the angular features (Equation (Equation 3)) as follows: $u=n_{s}$, $v=(p_{1}-p_{s})\times u$, w=u×v. The three angular features can be computed for a pair of points:(1)$$\alpha =v\times n_{s}$$
(2)$$\beta =\frac{p_{i}-p_{s}}{\left||p_{i}-p_{s}\right||}\times u$$
(3)$$\gamma =\arctan(t\times n_{t},u\times n_{s})$$

Let $N_{p_{s}}$ (Equation (Equation 4)) be the neighborhood of a query point $p_{s}$ in a given radius (*r*):(4)$$N_{p_{s}}=\left\{p_{i}|∥p_{s}-p_{i}∥<r\right\}$$

The simplified point feature histogram (SPFH) of a query point $p_{s}$ (Equation (Equation 5)) is the histogram of the angular features of all point pairs, in which the first point is the query point ($p_{s}$) and the second point is one of its neighbors:(5)$$\left\{\left(p_{s},p_{i}\right)|p_{i}\in N_{p_{s}}\right\}$$

In two steps (Equation (Equation 6)), we can obtain the FPFH feature of a query point $p_{s}$. First, calculate the SPFH for the query point $p_{s}$ itself. In the second step, calculate the SPFH for each point $p_{i}$ in the neighborhood ($p_{i}\in N_{p_{s}}$) (see Figure 3). After weighing the histogram of neighboring points with the distance, we get the final FPFH histogram:(6)$$FPFH\left(p_{s}\right)=SPFH\left(p_{s}\right)+\frac{1}{|N_{p_{s}}|}\sum_{i=1}^{|N_{p_{s}}|}\frac{1}{\omega_{i}}\times SPFH\left(p_{i}\right),$$
where $\omega_{i}$ is the distance between the query point $p_{s}$ and its neighbor $p_{i}$ ($p_{i}\in N_{p_{s}}$). The same steps are performed for Q.

### 2.3. Fitting of the Point Clouds, FGR then ICP

An approximate registration between $p_{i}$ and $q_{i}$ is performed to match the two point clouds, which requires the calculation of a point descriptor. FGR estimates a transformation that alights two point clouds together based on point correspondences between the two clouds. Correspondences are established by feature matching. These algorithms produce fewer tight matching results. The utility function represents the transformed source and target point clouds. Although the approach is faster than ICP, its performance could be better on large sample point clouds, which alone does not give valuable results. Therefore, a local refinement algorithm needs to refine the matching further [29]. Because ICP needs an initial transformation, it is used mainly in conjunction with FGR. Iterative matching uses the ICP method between the P and Q matrices. The ICP algorithm iterates through the following steps:1.**Determination of correspondences**First, we search for a correspondence set *C*, which contains matching index pairs. For each point in the P cloud, we search for its nearest neighbor from Q. Their indexes form a pair.2.**Transformation estimation**Find the center of mass (Equations (Equation 7) and (Equation 8)) of two sets of points:
(7)$$\mu_{Q}=\frac{1}{\left|C\right|}\times \sum_{(i,j)\in C}q_{i}$$
(8)$$\mu_{P}=\frac{1}{\left|C\right|}\times \sum_{(i,j)\in C}p_{j}$$Subtract the corresponding center of mass (Equations (Equation 9) and (Equation 10)) from every point:
(9)$$q_{i}^{\prime }=q_{i}-\mu_{Q}$$
(10)$$p_{j}^{\prime }=p_{j}-\mu_{P}$$Find a translation vector t and rotation matrix R (Equation (Equation 11)) that minimizes the sum of squared errors:
(11)$$E\left(R,t\right)=\sum_{(i,j)\in C}{∥q_{i}-Rp_{j}-t∥}^{2}$$This problem can be solved through singular value decomposition (SVD, Equation (Equation 12)), which is used to minimize the distance between the points [30]. We compute the cross-covariance matrix:
(12)$$W=\sum_{(i,j)\in C}q_{i}^{\prime }{p_{j}^{\prime }}^{T}$$We then use the SVD (Equation (Equation 13)) to decompose:
(13)$$W=UDV^{T},$$
where U, V are 3×3 rotation matrices.If rank W is 3, the parameters minimizing E(R,t) are unique and given by Equations (Equation 14) and (Equation 15):
(14)$$R=UV^{T}$$
(15)$$t=\mu_{Q}-R\mu_{P}$$3.**Transformation**Finally, the translated and rotated point (in Equation (Equation 16)) can be obtained:
(16)$$\bar{p_{j}}\leftarrow R\left(p_{j}-\mu_{P}\right)+\mu_{Q}$$4.**Iteration**Steps 1–3 are repeated, refining the transformation itself, getting closer to the overlap between the two point clouds so that the clouds fit together better.

After ICP, we get a final transformation that can be used to transform the point cloud. Now that the two point clouds have aligned, the distance between them can be calculated.

### 2.4. Clustering Based on Density

After the ICP, we can calculate the distance between two point clouds using the method *compute point cloud distance* provided by *Open3D*. It calculates the distance to the nearest point in the target point cloud for each source point cloud. We use Euclidean distance (Equation (Equation 17)) to calculate the distance:(17)$${\mathrm{NN}}_{Q}\left(\bar{p}\right)=\left|\right|\bar{p}-{q||}_{2}\;|\;\bar{p}\in \bar{P}\wedge q\in Q\wedge q$$
where ${\mathrm{NN}}_{Q}\left(\bar{p}\right)$ returns with the nearest point to $\bar{p}$ from Q.

The values will approach zero where the two clouds are nearly identical. However, the distance will increase if a discrepancy or a welding error is valid. For the distance values, a threshold is set to select the few thousand points that are “furthest” apart in terms of the position of the two point clouds. This can be a percentage threshold or a specific distance. The ${\mathrm{NN}}_{Q}^{10}\left(\bar{p}\right)$ is clustered as the top ten percent of ${\mathrm{NN}}_{Q}\left(\bar{p}\right)$. Because the location of these points varies with density, they are suitable for density-based analysis. Thus, the (sorted) distance values are considered the input dataset for the density-based algorithm.

Clustering is based on distance or similarity measures (e.g., density). The DBSCAN algorithm is a traditional density-based clustering method, currently presented in a *OPEN3D* Python environment. One of the key advantages of DBSCAN is that it does not require the user to specify the number of clusters. DBSCAN identifies points in a dataset in high-density regions and marks them as *core points*, as shown in black in Figure 4. These *core points* are then used to define clusters, data regions densely populated with points. It then expands outward from these core points, adding points within a certain distance (called the *eps* parameter) to the cluster. *Core points* contain at least *minpoints* points in their *eps* neighborhood. *Border points* (shown in orange in Figure 4) does not contain enough points in their neighborhood, but they fall in the neighborhood of some *core points* [31]. Points that are not within the *eps* distance of a *core point* are considered *noise* (red in Figure 4) and are not included in the clusters [32].

In our case, the *eps* value is related to the pcr value (Equation (Equation 18)), which has the magnitude of the arithmetic mean of the distance between each point in the point cloud and its neighbors (where the quotient has the number defined for sampling):(18)$$\mathrm{pcr}\left(Q\right)=\sum_{\bar{p}\in Q}{\mathrm{NN}}_{Q}\left(\bar{p}\right)/N$$

The *minpoints* parameter of the DBSCAN is the minimum number of points per cluster required for a point to be considered a dense region or valid cluster. The cluster number indicates the minimum number of points a cluster can consist of.

After clustering, we can select the cluster that seems to be the problem and use the root-mean-square error (RMSE) formula to determine its distance from the master cloud. RMSE is calculated by following formula:(19)$$RMSE=\sqrt{\sum_{i=1}^{N}\frac{{\left(p_{i}^{\prime }-q_{i}^{\prime }\right)}^{2}}{N}}$$
where *i* is the estimated time series, and the denominator is the sampling volume. For each point in the identified clusters (errors), we examine how far it is from its nearest neighbor in the reference. The higher RMSE value means a more problematic cluster.

## 3. Application of the Developed Clustering Method for Welding Defects Identification

In this section, we will prove the applicability of the developed framework on generated workpieces (see Section 3.1) and an actual physical workpiece (see Section 3.2). We will show how we can detect welding defects based on the proposed framework in both cases. Welding technology standards are used to assess the variation in weld defects, which can be used to accurately determine the adequacy of welds. One of these standards is *ISO 6520-1:2007*, which deals with the main classification of defects and their precise description. The six main groups of defects are described in tabular form, each with an example (see Table 1). The other standard that shows in detail the individual seam deviations and their limits is *ISO 5817:2014*. The welding defects are categorized based on whether they can be detected with the developed ML-based solution. The standard includes limits in three quality categories or indicates acceptability by the terms *not permitted* and *permitted*. The acceptability of a welded joint is based on the visible absence of defects in the weld and the adequacy of all the tests described in the relevant procedure test standard. All types of welds-related deviations have a “catalog number” or classification, given in Table 2 with limits. The *number of imperfections* is the number of the deviation recorded in the standard. We calculated this for A5 weld sizes.

We applied the following software and hardware elements for the tests of the developed framework:*3D scanning:* HP 3D Structured Light Scanner 5 Pro Edition scanner (0.05 mm maximum accuracy) with software (vers.: 5.2.0.790), HP Development Company, L.P., Germany*CAD models:* Solid Edge Academic Edition (vers.:221.00.00.114),*Point clouds to creating:* Meshlab (fpv2020.12),*Algorithm environment:* Python (vers.: 3.7.7),*Visualization field:* Open3D (vers.: 21.1.3).

### 3.1. Performing Tests on Generated Workpieces

After fitting the CAD point clouds, the results of density-based clustering (from the density characteristic (radius=15×pcr) and a minimum number of points per cluster is 100) are shown in Figure 5. The generated defects, except for two (crack and end crater), form a separate cluster and are visible. The evaluation is defined as different marginal deviations in the three quality categories (*B*, *C*, *D*). As you move down to BCD, the standard becomes more permissive; for example, the height of the excessive convexity can be up to 3 mm for category *B*, 4 mm for category *C*, and up to 5 mm for category *D*.

In the case of cracking, the defects generated do not show up as separate clusters, even when moving towards the more permissive direction of the standard. The other main groups of defects (solid inclusion, lack of fusion, imperfect shape, and miscellaneous imperfection) are detectable by clusters (see Table 2):**Crack** (number of imperfection **100**): For the test, a “brick body cavity” of 0.5mm×0.2mm×0.1mm was created in the CAD model (Figure 5a). Regardless of the defect type, the system detects the same percentage of the parts remaining outside the scattered areas together with the cavitation defect cluster for both defect clusters (225,000 points for 90%). The least sensitive group for the test, due to the size difference (100 points/250,000 points), small and fragmented clusters are not visible to the system, and the test cannot detect clusters with such low scores.**Excessive convexity** (number of imperfection **503**): (Figure 5b).In case of **Imperfect shape** (number of main imperfection **500**):
-**Insufficient throat thickness** (*Smaller root size*—number of imperfection **5213**): detectable in categories *C* and *D*.-**Excessive throat thickness** (*Larger root size*—number of imperfection **5214**): (Figure 5c) detectable in categories *C* and *D*.

Clusters with roughly the same number of points for both error groups can respond differently to the test because they have different geometric properties (for the case of excessive convexity, a normal vector plus the reference). Above 90% (a score of approximately 25,000), the two groups separate in terms of cluster scores, whereas for bump-like defects, defects can still be detected with a minimal decrease; for other imperfection groups, after a marked decrease, no defect-presenting cluster can be detected at 99%.

**End crater pipe** (number of imperfection **5012**) (Figure 5d) and **Intermittent undercut** (number of imperfection **2025**) (Figure 5e): In these cases, minus normal vector errors proportional to the reference point cloud are detected. The number of points in the cluster detected as an error decreases steadily, and at 90%, the cluster product of the generated error contains 5700 points. The linear decrease is also true for the error group of the end craters, but the number of points in the cluster clouds is smaller, from 288 to 176.**Excessive asymmetry of fillet weld** (number of imperfection **512**) and dimension: The most sensitive group of errors to analyze, the diversity of errors means that detection is roughly 90% successful (Figure 5f). At this optimum percentage, the test detects five clusters (all the errors represented).

In conclusion, density-based clustering cannot detect crack and end crater defects due to their small size. Based on the location and extent of the clusters, the asymmetry defect shows two distinct clusters on the seam, with excessive convexity and intermittent undercut showing one cluster each along the entire length of the seam.

### 3.2. Performing Tests on Real Physical Workpieces

We used a *Telwin Inverpulse 320* welding machine to create welded samples: 8−8 specimens, 50 mm long, with a5 welds (dimension of the height dimension of the triangle enclosing the corner weld) were T-welded. We will show the lack of cutting accuracy required to produce the part, and various burn-in problems can also be detected by examining the joints. The defective test piece was fitted to a reference workpiece (validated best workpiece). As shown in Figure 6, in addition to the differences in weld size, differences in geometrical size also appeared separately (workpiece corners or deviations from planes). It is important to note that, whereas in Figure 6, the red colors represent the furthest points from the reference workpiece, in Figure 7, red is just one of the colors of the clusters sign.

We derived the value of eps from the density characteristic (15×pcr). According to this radius, we have checked whether each point has a neighbor. The minimum number of points per cluster has been set to 100 in this case, based on the number of dimensions present in the dataset and the number of dimensions in the data [34]. The top 10% of the distances are obtained after the fits are clustered: the center of clusters and size distributions exhibit characteristics typical of the types of weld defect. As many clusters as detectable, as many errors are identified.

The testing process is similar to that described for generated workpieces. However, after DBSCAN, not only a “perfect” defect is displayed. Therefore, the defect clusters were sorted by their RMSE value (most significant value, upfront) and labeled by their center of mass location on the workpiece (brown point—maximum RMSE value). As shown in Figure 7, two of the center of mass points fall on welding seams (brown—down, blue—up), but based on the sorted RMSE values, only the lower cluster requires further investigation. The long red cluster does not exceed the tolerance values. The center of mass of the other visible clusters is located inside the part.

Based on the RMSE values and the center of gravity values, whether the defect is currently an excessive convexity defect (above the allowable value) or an intermittent undercut defect (below the value) can be further investigated. The cluster indicates a weld defect if the marking falls on a weld. In this case, the product must be repaired or scrapped. The values of the clusters to be listed are shown in Table 3. Typical weld defects are excessive convexity defects, where one of the centers of mass of the defective clusters has a value beyond the tolerance, higher than the weld line height of the defect-free part. The tolerance mentions three categories (*B*, *C*, *D*). As noted, the actual size is outside the *B* tolerance category in one case.

The furthest values for fitting generated CAD models were 4.803 mm (*B*), 3.939 mm (*C*), and 2.936 mm (*D*). The values are close to the standard (5–4–3 mm).

In conducting these experiments, we found the following:It is inefficient for the framework of the CAD model to contain errors of too small a size; below a score of 200 (out of 250,000), no error is detectable. (Crack and end crater defects cannot be detected.)Sub-surface defects cannot be detected due to weld seam gassing, and additional investigations are needed.Comparing the image of the CAD model with the image of the welded workpiece is unfortunate due to geometric differences, as differences not related to the weld scan are also considered errors. Dimensional differences caused by inaccurate cuts generate huge clusters.A glossy, transparent surface or all-black-and-white color will spoil the expected scanning result. However, environmental preparation is also essential (from the projector light, no brighter light in the room).Care had to be taken to fine-tune the scan in the setup, but it is highly hardware-dependent. Generating a 3D model from more than six (rotation angles less than 60 degrees) matching images makes the process slow.

## 4. Conclusions

This study presents a method for classifying weld defects in corner welded joints using imaging techniques and data clustering. The developed framework employs a results-oriented approach to visualize defects using CAD, with point clouds prepared for coarse (FGR) and fine (ICP) matching by point features (PFH). Density-based clustering (DBSCAN) is applied to the sorted distance between point pairs, resulting in several clusters. The location of these clusters is then clarified by sorting and marking them on the workpiece. The excessive convexity defect is presented as an example, with RMSE values of the clusters compared to the standard. Clusters exceeding the limit value are marked by their center of mass, which can be used to check whether the cluster with the critical value is located on a weld seam. The generated and physical workpieces demonstrate the applicability of the developed framework.

The limitation of the developed framework depends on the 3D scanning technology. The quality of the scan is strongly influenced by the physical conditions: the settings used in the preparation; the camera position away from the subject; the angle of the camera to the horizontal or camera to the projector; the light conditions in the scanning room. In addition, the machine vision methods can lack robustness when faced with variations in the manufacturing process or imaging conditions [35]. Using the DBSCAN algorithm parameters eps=100 can cause a failed defect detection for extremely small (0.5 mm × 0.2 mm × 0.1 mm) weld defects (e.g., cracks), as less than 100 points are generated on the defect cluster surface out of 250,000. Further reducing the value of eps, however, thousands of clusters will be misclassified. It is important to note that for generated CAD models if a fault passes through a full cross-section, some bias occurs in the calculation of its maximum distance. It is emphasized that this method alone cannot be used for a complete weld analysis, but it can speed up and automate the process. Based on standardized tests, it is concluded most of the listed defects can be detected, with some limitations. The center-of-mass coordinates of point sets can be used for graphical applications, allowing for the visualization of the defect location for quality control purposes, and potentially launching further research in this area.

The proposed method can be integrated into digital production processes, as it enables the automation of the inspection process. This automation can either support or completely replace human decision-making. However, it may result in incorrect decisions caused by surface errors during rendering. To overcome this limitation, future work could consider combining multiple methods and utilizing advanced convolutional neural techniques as a potential solution.

## Figures and Tables

**Figure 1 sensors-23-02503-f001:**
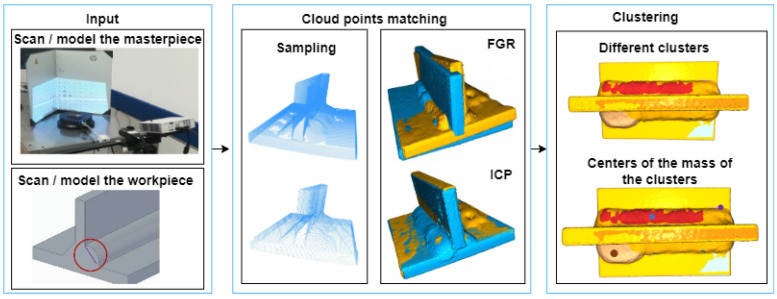
Steps of the error analysis method: *Input:* the masterpiece and the actual workpiece are scanned or modeled (CAD) as an input. *Cloud point matching:* sampling and fitting the point clouds by fast global registration (FGR) and technique of iterative closest point (ICP). *Clustering:* density-based clustering to get the centers of the potential faults.

**Figure 2 sensors-23-02503-f002:**
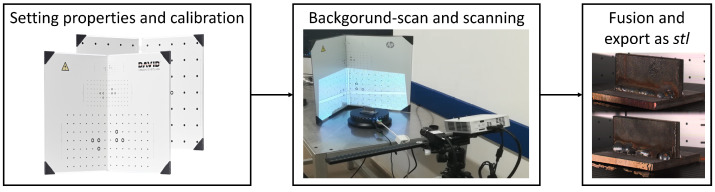
The 3D scanning process: (1) calibration of the tray with recommended properties; (2) 3D scanning, camera position 6–8 cm to the left of the projector, camera angle to guide rail 100–102 degrees, tripod position top view of the subject of no more than 30 degrees) scanning with 6–8 shots; (3) point clouds fusion and extraction.

**Figure 3 sensors-23-02503-f003:**
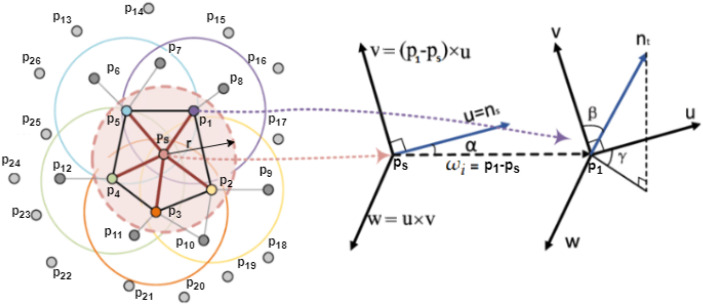
Calculating of FPFH. The query point is $p_{s}$ and its neighbors are: $p_{1}$, $p_{2}$, $p_{3}$, $p_{4}$, $p_{5}$. The angular features are calculated for every point pair connected with a black line.

**Figure 4 sensors-23-02503-f004:**
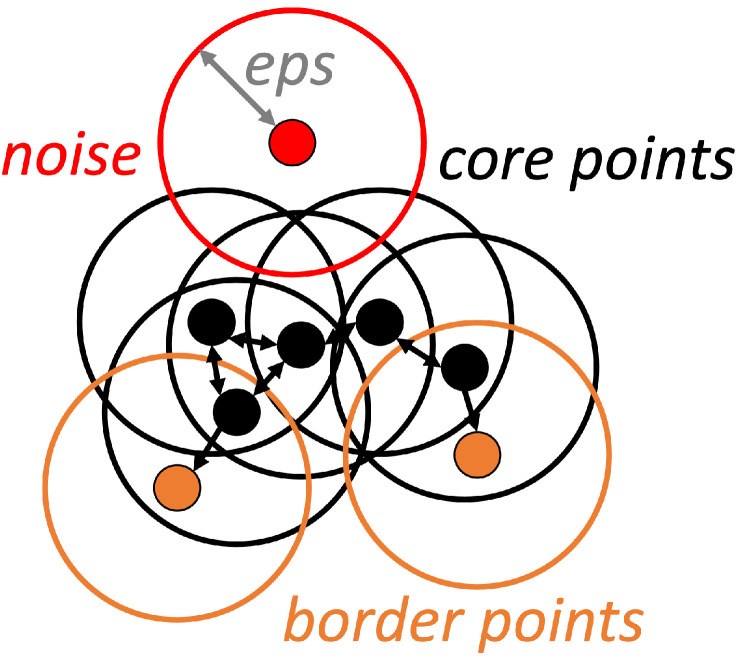
Parameters and process of DBSCAN. DBSCAN begins by selecting a point randomly and checks whether the selected point is a *core point* (if it contains at least *minpoints* number of minimum points in its *eps*-neighborhood). It assigns each non-core point to the nearest cluster, or otherwise assigns it to *noise*.

**Figure 5 sensors-23-02503-f005:**
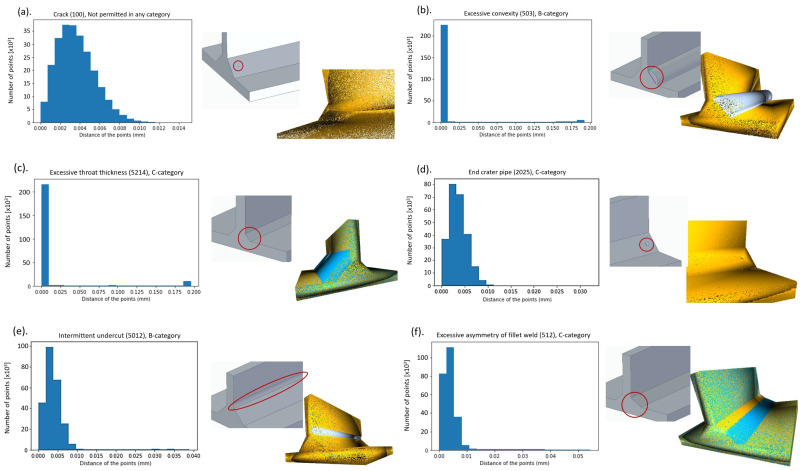
Exemplary distributions—CAD models of welding deviations.

**Figure 6 sensors-23-02503-f006:**
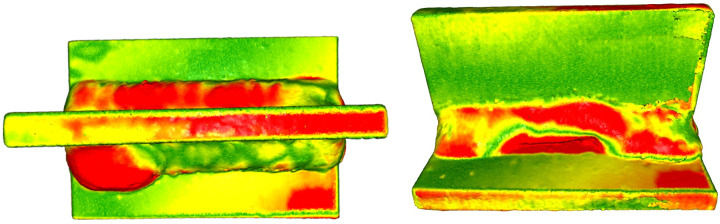
Welding defects: excessive convexity defect (**left**) and intermittent undercut defect (**right**), based on color temperature.

**Figure 7 sensors-23-02503-f007:**
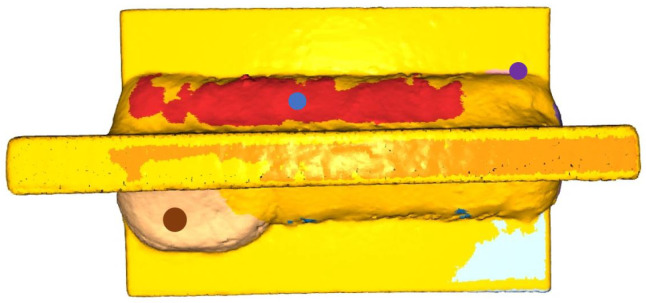
Excessive convexity defect with centers of the mass from the top view. The index of red is outside of the tolerance (3 mm) category *B*.

**Table 1 sensors-23-02503-t001:** Main imperfections and examples (according to ISO 6520-1 [33]). List of the main groups of imperfections, their corresponding group numbers and descriptions, as well as an example of each imperfection with an accompanying figure are included.

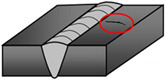	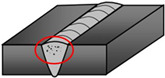	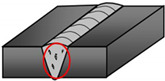
**100**—**Crack**, example: transverse crack in the parent material (1024)	**200**—**Cavity**, example: gas cavity (2011)	**300**—**Solid inclusion**, example: slag inclusion (301)
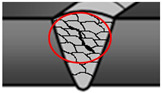	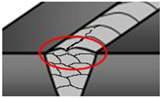	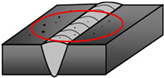
**400**—**Lack of fusion and penetration**, example: lack of inter-run fusion (4012)	**500**—**mperfect shape**, example: inter-run undercut (5014)	**600**—**Miscellaneous imperfection**, example: spatter (602)

**Table 2 sensors-23-02503-t002:** Welding defects in fillet welds (according to ISO 5817) [26].

Limit Values			
**Imperfections**	**B**	**C**	**D**
Crack (100) (Figure 5a)	not permitted	not permitted	not permitted
Excessive convexity (503) (Figure 5b)	3 mm	4 mm	5 mm
Insufficient (5213) (Figure 5c)	not permitted	1 mm	2 mm
Excessive throat thickness (5214) (Figure 5c)	3 mm	4 mm	5 mm
Endcrater pipe (2025) (Figure 5d)	not permitted	1 mm	2 mm
Intermittent undercut (5012) (Figure 5e)	0.5 mm	0.5 mm	1 mm
Excessive asymmetry of fillet weld (512) (Figure 5f)	2.25 mm	2.75 mm	3 mm

**Table 3 sensors-23-02503-t003:** The values of the clusters. The text colors are matched with Figure 7.

Cluster ID	Number of Points	RMSE	Center of the Mass in Case of Excessive Convexity Defect
0	3824	3.165	127.278, −57.460, 146.059
1	8433	2.089	160.334, −55.132, 107.931
2	6058	1.742	114.276, −64.487, 146.245
3	618	1.487	151.439, −61.352, 104.103
4	3741	1.106	144.248, −31.916, 121.838
5	172	0.832	130.808, −57.191, 124.218

## Data Availability

Publicly available datasets were analyzed in this study. This data can be found here: https://github.com/abonyilab/3D-scanner-data (accessed on 28 December 2022).

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
