# Peer review of "3D Scanner-Based Identification of Welding Defects—Clustering the Results of Point Cloud Alignment"

_sensors, 2023, doi:10.3390/s23052503_

Round 1

Reviewer 1 Report

The scientific paper "3D scanner-based identification of welding defects – Clustering the results of point cloud alignment” proposes a method for examining and evaluating of welding defects by using 3D scanners where welding errors are detected by comparing obtained point clouds with density-based clustering.

I can make the following considerations as the Authors may improve their manuscript by following these suggestions:

Abstract does not contain any quantitative data or clear results, only method.

Introduction section is thin in my opinion. Authors should do more diligence on more comprehensive application of 3D scanning technologies and other methods/procedures for examining and evaluating of welds and welding defects from other relevant research articles.

Abbreviation “CAD” should be elaborated where it was first mentioned in the manuscript.

In the Introduction section, the following sentence “Structured light scanners work similarly to laser scanners, but the light source is not a line laser but a high-resolution projector that projects a raster mesh onto the surface of the object.” (line 34 to 36)

Replace “a raster mesh” with “coded patterns” as it depicts proper terminology.

One would argue with Authors for their statement that structured light 3D scanners are more accurate than their laser counterparts by citing reference [8]. This vastly depends on many factors like their application, materials properties, surface quality, scanning conditions, etc. I advise Authors to better clarify their statement and support it with more high-end papers.

“DBSCAN (Density-based spatial clustering of applications with noise)” (line 46) – reverse the order to “Density-based spatial clustering of applications with noise (DBSCAN)”

“ICP (iterative closest point)” (line 71) – remove the text in round brackets, as ICP abbreviation has been explained above already. Same goes for “The FGR (fast global registration) filters…” (line 138). Check this throughout entire manuscript.

Workflow shown on Figure 1 is somewhat confusing, and hard to follow. It’s difficult for make out where developed method starts and where it ends. Please clarify it better.

“The structured light camera system creates the model. The 3D scanner creates pictures and makes the 3D point clouds.” (line 84 to 85) - Terminology is not correct in these two sentences. Briefly explained, structured light 3D scanners reconstruct the raw scan of the object being digitized which is then converted into high-density point cloud. After that, the point cloud is converted to mesh 3D model (depending on the software). Please correct this.

Sentence “The purpose of the analysis has to obtain….” Replace word “has” to “was”.

The section below Figure 1 about obtaining CAD files is confusing. CAD parts with defects and a reference CAD workpiece are modeled from what exactly? Based on what? Are CAD models modeled according to the scanned welds, standards, or..? If they are modeled from scanned data, how do Authors accompany the fact that the CAD model has to be modeled according to the scanned weld and doesn’t that involve a certain degree of dependent data due to the fact the CAD files are derived from these scans? Is there a better way to do this? This entire section is confusing, please elaborate this further.

Sentence “For the fitting, important to detail the descriptors of the point clouds.” (line 112) is confusing, elaborate it better.

More information about the 3D scanner used is needed. There is no information about its manufacturer, model number, etc.. Also, information about its accuracy is missing, which should be added considering the scope of this manuscript.

Same goes for software used for CAD modeling.

Add the sentence “RMSE (root-mean-square error) is calculated by following formula:” or something in that context to support Equation (19).

The following sentence is repeated in both Introduction and Application of the developed clustering method for welding defects identification sections:

“The standard includes limits in three quality categories and indicates acceptability by the terms not permitted and permitted. The acceptability of a welded joint is based on the apparent absence of weld defects in the weld and compliance with all test types described in the relevant procedure test standard.” Its better to reiterate the second sentence, or just remove it.

Make Table 1 a little bit more compact, and also try to avoid cutting the last sentence above the Table 1 on page 9. Same goes for last sentence on page 10 (Sub-section 3.2. Performing tests on real physical workpieces) and page 11.

I believe that the numerical values obtained in main groups of defects (Sub-section 3.1. Performing tests on generated workpieces) would be clearer if presented in a table.

Move figures 6 and 7 below text where they are first mentioned (below line 303).

Also, in context of figure 6, Authors should perform CAD-Inspection analysis which would improve manuscript better, as it will show actual dimensional deviations of scans from CAD at those areas that are represented with the furthest points.

Move Table 2 below text where it was first mentioned (below line 323).

Reviewer 2 Report

Please read the attachment. Thank you.

Round 2

Reviewer 1 Report

I can see that the Authors did acknowledge most of my remarks for their manuscript, and improved it substantially.

Few small corrections should be performed, but do not require additional revision on my part as they are minor:

“3D scanning is a precise and fast non-contact measurement process…..” (line 51) – replace “precise” with “accurate” as they have a different meaning in the context of performance of 3D scanning systems.

In Figure 1, replace the word “draw” to “model” if Authors are referring to the acquisition of CAD model via CAD software. Same goes for the description of Figure 1 “Steps of the error analysis method: Input: the masterpiece and the actual workpiece are scanning or drawing (CAD) as an input.” – replace “drawing” with “modeling”.

Add “3D” to the description of Figure 2: “2) 3D scanning, camera position 6 - 8 cm to the left….”

Move the following bullets above Table 1: (lines 283 and 284)

Algorithm environment: Python (vers.: 3.7.7),

Visualization field: Open3D (vers.: 21.1.3),

Use spacing or comma for the following: “(0.5mm0.2mm0.1mm)” (line 401)

Wishing Authors all the best in their future work.

Reviewer 2 Report

Dear Editor and Authors: 

Thank you for providing the point-to-point response.

The authors have carefully corrected and answered my comments and questions. The main script is ready for publishing. The reviewer strongly suggests it be accepted for publication. 

Thank you for reading. 

Sincerely yours, 

The Reviewer.

Author Response

Thank you very much for all your support and valuable suggestions.